# Limitations and opportunities of smallholders' practical knowledge when dealing with pig health issues in northern Uganda

Anna Arvidsson[1]*, Klara Fischer[1], Erika Chenais[2], Juliet Kiguli[3], Susanna Sternberg-Lewerin[4], Karl Ståhl[2]

**1** Department of Urban and Rural Development, Swedish University of Agricultural Sciences, Uppsala, Sweden, **2** Department of Epidemiology and Disease Control, National Veterinary Institute (SVA), Uppsala, Sweden, **3** Department of Community Health and Behavioural Sciences, Makerere University, Kampala, Uganda, **4** Department of Biomedical Science and Veterinary Public Health, Swedish University of Agricultural Sciences, Uppsala, Sweden

\* anna.arvidsson@slu.se

**Data Availability Statement:** The datasets cannot be shared publicly, as they contain both identifying and plausible sensitive participant information. The participants gave their consent with the

## Abstract

Pig production has a short history in Uganda. The majority of pigs are kept by smallholder farmers in rural areas where access to veterinary services is limited, and pig keeping has been suggested as a potential pathway out of poverty for smallholders. Previous research has identified the disease of African swine fever (ASF) as a major threat, causing high mortalities in pigs. With no available cure or vaccine, the only option is to implement biosecurity measures, i.e. strategies that prevent the spread of ASF. This paper draws on data from four months of ethnographic fieldwork in rural northern Uganda. Combining methods of participant observation, semi-structured interviews, focus group discussions and a survey, the aim was to improve understanding of smallholders' perceptions and responses to pig health issues such as ASF. Applying the concept of practical knowledge, this paper analyses the potential and limitations of smallholders' practice-based knowledge as a means of dealing with pig health issues. The results show that while pigs were appreciated locally for providing an income, many informants found it difficult to deal with pig diseases effectively. Consequently, informants commonly expressed a need for other kinds of knowledge in their pig production, indicating that veterinary advice can play an important role in reducing the negative impact of pig health issues. For animal health provision to have relevance in this context, however, veterinary practitioners must pay close attention to smallholders' priorities and ways of knowing in their livestock keeping. Results further show that pig health issues made some informants abandon pig production altogether. To enhance the potential of pig production as a poverty mitigation strategy in Uganda, research and policy need to focus on ways of bettering general conditions for smallholder pig keeping, including improving the quality of and access to veterinary services in rural areas.

understanding that their responses would remain confidential. With this said, researchers can request access to anonymized data by contacting the Swedish National Veterinary Institute (SVA) (contact via sva@sva.se). All other relevant data are available in the results section of the manuscript, including results from individual interviews and focus group discussions.

**Funding:** This study was funded by Vetenskapsrådet (https://www.vr.se/), Award Number 2017-05518. The funder had no role in study design, data collection and analysis, decision to publish, or preparation of the manuscript.

**Competing interests:** The authors have declared that no competing interests exist.

## Introduction

This paper reflects a growing interest in research into the social and cultural aspects of animal diseases [1]. Drawing on data from ethnographic fieldwork in northern Uganda, this study explored smallholder farmers' perceptions and responses to pig diseases in general, and the disease of African swine fever (ASF) in particular.

ASF is a viral disease that affects pigs and can lead to severe clinical disease and death [2]. It is endemic in Uganda and has a significant negative impact on the country's pig production and farmers' financial situation [3]. Infected pigs typically develop clinical signs such as a loss of appetite, high fever and haemorrhages leading to skin colour changes [4]. In most cases, the infected pigs die within a few days. Although ASF has been known for more than 100 years [first described in Montgomery 1921, see 5], there is still no vaccine or cure for it. Instead, its spread can only be prevented by basic biosecurity measures, such as avoiding direct and indirect contact between naïve pigs and infectious pigs and materials. Previous research indicates that it is particularly difficult to implement biosecurity measures successfully in the smallholder context, and points to smallholders' limited financial means as well as lack of access to veterinary support as key challenges [6]. Overall, previous veterinary and social science research in Uganda has identified a need for more locally adapted biosecurity measures to prevent the spread of ASF and reduce its negative impacts in poverty-constrained smallholder contexts [7, 8].

This study was conducted in northern Uganda, a part of the country still recovering from an extended period of armed conflict that took place between 1986 and 2006 [9, 10]. During the conflict, the majority of people in the north were forced to stay in so-called internally displaced persons (IDP) camps, in which access to agricultural land was highly restricted [10]. Many also lost their livestock during this time and were consequently left poorer [9, 10]. When the armed conflict ceased and it was safe to return to their former home villages, smallholders were slowly able to resume cultivation and livestock keeping [9]. In this context, and with the aim of reducing poverty and rebuilding rural economies that had been severely affected by the long-term conflict, among other initiatives the government and donors have promoted pig production [11]. There has been growing recognition of the benefits of pig production and it is now a fairly common livelihood activity in the study area. This is also reflected in the increase in the number of pigs in northern Uganda, where the pig population has grown from about 100,000 in 2002 to 350,000 in 2008 according to the latest national livestock census [12]. Previous studies illustrate that Ugandan smallholders often have inadequate access to existing veterinary services [13–15], therefore they are largely left to their own knowledge and locally available resources to deal with animal health issues in their livestock production. This is part of a wider tendency in sub-Saharan Africa, where veterinary services and advice are often being modelled to serve commercial and large-scale farmers [16–19].

Complementing previous research, the present paper explores how smallholders conceptualise animal disease in general and ASF in particular. Its findings are expected to provide important information to policy-makers with regard to communicating successfully about ASF in the smallholder context.

## Conceptual framework

Over the years, scholars have used a wide variety of concepts to theorise farmers' knowledge. Earlier preoccupations with defining and emphasising binaries between scientific knowledge and farmers' local knowledge have declined, and the longstanding term of "traditional" knowledge has increasingly been abandoned due to problematic connotations of "backwardness" in relation to so-called "modern Western science" [20, 21]. A key focus of more recent

publications about smallholders' understanding of livestock health has been on concepts of "hybridity" and "pluralism". They reveal how smallholders' knowledge is essentially adaptive and experimental, and draws from multiple sources, including practical knowledge from within the community and biomedical knowledge from external veterinary advisors [22–24].

Common to past and present conceptualisations of smallholder farmers' knowledge is the acknowledgment that smallholders have in-depth knowledge of their local environments. This plays a critical role in providing the most appropriate local solutions when dealing with problems in farming [20, 25, 26], including ways to deal with animal disease in the local context [see, for example, 22, 24, 27, 28]. Many authors who have written about smallholders' agricultural knowledge emphasise that it is adapted to the local context and complexity [20, 25, 26], is largely tacit, and is often passed on through demonstration, observation and practice [24, 25, 29, 30]. This practical knowledge can further be understood as evolving through a process of constant interpretation and evaluation, where fine-tuning of methods and the search for better solutions often develop during ongoing discussions and knowledge-sharing with other community members [25]. This kind of local knowledge could also be described in terms of "craftmanship", closely related to skills, in which elements such as commitment and passion are suggested to influence the ability of smallholders to breed healthy animals and achieve high crop yields [25]. Scott [20] uses the ancient Greek word "métis" to refer to smallholders' practice-based, situated knowledge. He also emphasises that métis is commonly sufficiently precise to serve its purpose, but no more than that. This is because the purpose of local, practical knowledge (or métis) is to solve concrete problems at hand, rather than contribute to a generalised body of abstract and precise knowledge about an issue (as in science) [20].

While the smallholders in this study had extensive experience of livestock keeping, they also reported not having sufficient competence to deal with animal disease adequately, particularly disease in pigs. This indicates that a combination of smallholders' and veterinary practitioners' knowledge is needed to identify adequate ways of treating livestock disease [see also 22, 26]. As mentioned above, other studies have recognised that many smallholders use a combination of what they learn from veterinary practitioners and their local practical knowledge in their livestock production, illustrating how knowledge systems tend to overlap in everyday practice [22, 24, 27, 31]. Current research also indicates that agricultural policy and veterinary practice often fail to account for and understand the value and purpose of practical knowledge sufficiently [20, 24, 27, 28, 32]. Indeed, there is a widespread tendency in agricultural development to prioritise formal scientific knowledge over local practical knowledge, and assume that people relying on practical knowledge are in need of "improvement" based on scientific advice from outside experts [13, 20, 25, 32, 33]. It has been shown that this insensitivity to local knowledges and practices among policy and advisory services is an important reason for suboptimal use of veterinary advice in the local context, resulting in suboptimal treatment of disease [34].

In order to improve understanding of how smallholders' local practical knowledge and veterinary knowledge can fruitfully be combined to find ways of dealing with ASF and other animal health issues in the local context, this study began with smallholders' ways of knowing and acting on animal disease. This approach of building on and strengthening smallholders' existing local knowledge has both been suggested in relation to dealing with ASF in Uganda [35], and proven crucial for achieving effective disease control in other sub-Saharan contexts [36]. In other words, for external advice from scientists and veterinary actors to be relevant in the local context, it is first necessary to understand the aims and methods of smallholders' practical knowledge.

## Materials and methods

An ethnographic approach was chosen on the basis of its potential to provide rich insights into smallholders' experiences and practices in relation to animal health issues, as well as the broader context in which their livestock production and ways of knowing are embedded.

### Study setting

Data were collected in two villages in Nwoya district, Acholi sub-region, northern Uganda. The district is predominantly rural and has a population of approximately 130,000 people [37]. The climate is tropical, with a rainy season stretching from April to November and a dry season from December to March. The authors' previous research and key contacts in the study villages and documented reports of ASF outbreaks were the factors that determined the choice of study area.

The vast majority of data were collected in what is referred to here as "village A", where the first author stayed with a Ugandan family during fieldwork. Complementary data were collected in the home village of one of the field assistants, referred to here as "village B". Smallholders in both villages commonly divide their time between crop and livestock production, and some of them also run small-scale business enterprises on the side [as described previously in 38]. The village centres are the locations of several of these businesses, including local bars, food joints and hair salons. In village A, two health clinics offer minor treatments to villagers, while access to pharmaceuticals for livestock and formal livestock markets requires travel to a nearby town or to the nearest city of Gulu. The main road connecting the city of Gulu and the capital Kampala can be reached by a 30 to 40-minute motorbike ride from the centre of village A. The distance between village A and B is about 30 kilometres. Village B is located alongside the tarred main road and served as an IDP camp during the most recent conflict. The size and population of village B are slightly larger than village A, and the range of services more comprehensive. The main livestock reared in villages A and B are poultry, goats and pigs. While cattle rearing is very rare in village A, it is more common in village B due to the availability of community grazing land. In the fields for crop production, often located close to smallholders' homes, rice, groundnuts, cassava, sesame and other crops are commonly produced.

### Data collection

Ethnographic fieldwork was carried out by the first author from September to December 2019, and smallholders in approximately 70 households were interviewed [see also 38]. In this paper, specific attention is paid to the responses related to livestock production, and more specifically to animal health issues in pigs. Alongside the semi-structured interviews, the first author observed and participated in smallholders' daily lives and farming practices. In addition to interviews and participant observation, six focus group discussions with a total of 43 smallholders were organised. The overarching aim of these discussions was to contribute greater understanding of the participants' views on the challenges related to livestock. Participants for the focus group discussions were chosen using the selection criteria of previous experience of livestock production and being over the age of 18. One of the field assistants guided each discussion by asking open-ended questions, and the other translated from Luo into English. The first author took detailed notes and intervened when clarification or follow-up questions were deemed appropriate. Since mixed groups with men and women risked being dominated by the perspectives of male participants, two groups were women only. All except one focus group discussion included a ranking exercise at the end of the discussion. Participants were asked to rank the challenges that had been mentioned during the discussion, with the purpose of capturing the perceived magnitude of each challenge. In two of the groups, the participants choose

to discuss challenges with pig production, while the other groups decided to focus on challenges with goat and poultry production, with which they had more experience.

The names of animal diseases in this paper are taken from the field assistants' English translations of smallholders' responses in Luo. The first author discussed these translations with the assistants to ensure a translation that was close to the smallholders' intended meanings, while avoiding forcing smallholders' categorisation of diseases or problems in their livestock production into specific disease names in English if that was not the smallholders' initial meaning.

In the final stage of fieldwork, a survey was designed with the aim of cross-checking and quantifying the qualitative findings [39]. The survey focused on smallholders' perceptions of problems with livestock keeping and access to veterinary services. In the survey, smallholders were asked to rank the key challenges in livestock production that had previously been mentioned in interviews and focus group discussions. The survey was delivered by one of the field assistants, trained by the first author, who interviewed a total of 101 smallholders (16 from village A and 85 from village B) in Luo and wrote down responses in English. A mix of purposive and convenience sampling strategies were applied when selecting informants. The criteria that smallholders had to meet to be selected for the survey were that they were adults with previous knowledge of livestock production who were at home at the time of the field assistant's visit. The predominant number of informants from village B was due to convenience because this was the field assistant's home village.

## Data analysis

Interviews and focus group discussions were not recorded, and thus not transcribed *verbatim*. Instead, the first author made detailed notes during and immediately after the interviews and focus group discussions. To avoid misunderstandings (thus ensuring the validity of the data) and identify potential gaps in the data, findings were frequently discussed with key informants and the field assistants throughout the fieldwork [39]. In addition, findings were discussed with the District Veterinary Officer (the person ultimately responsible for animal health in the district) as well as veterinary officers working in Nwoya district at the time of fieldwork in 2019. In this sense, the data analysis already started in the field. After the fieldwork, interview transcripts were imported into NVivo 12 (QSR International) and the first author continued the analysis by carefully rereading all notes as a way of becoming familiar with the material [40]. This close reading of the material was combined with coding, initially focusing on exploring potential connections and contradictions within the material. Broader themes around the studied smallholders' livestock production evolved at this stage of the coding, such as "pig keeping", "animal diseases" and "livestock advice". These broader themes were discussed with the co-authors, identifying potentially interesting aspects on which this paper could focus and exploring varied interpretations of the data. In relation to this process, the research questions for the paper became clearer and the relatively broad themes of the codes developed into narrower topics, for example "local treatment methods", "handling of dead pigs" and "syndromes that can be interpreted as ASF". Application of theoretical concepts (presented in the conceptual framework) enabled identification of aspects in the empirical material that might otherwise have been overlooked and made it possible to generalize through theory [41]. In other words, the analysis was both inductive and deductive.

Survey data were collected on paper questionnaires by one of the field assistants and later entered into a Microsoft Excel spreadsheet for data analysis by the first author. This helped provide an overview of the data and determine the minimum, maximum and average in the quantitative results.

### Ethical statement

This study was reviewed and approved by Makerere University, College of Health Sciences Research and Ethics Committee, under reference number 2019–062. Prior to participation, all the smallholders were provided information about the overarching aim and expected outcome of the study. They were also told that they could decline to take part of the study at any time and for any reason. Oral informed consent was given by all smallholders prior to participation. In accordance with the ethics approval (reference number 2019–062), the oral consent was documented in the written interview notes by the first author, who attended each interview, and witnessed by a field assistant. To protect the anonymity of the participants, the smallholders' names have been changed and the names of the study villages have been excluded.

## Results

While cattle, goats and poultry had been part of the everyday lives of the studied smallholders since childhood, pigs were introduced more recently. Several informants had their first interactions with pigs in IDP camps, and decided to invest in pigs when they returned to their villages about 15 years ago. While pigs were indeed present and reared in the study area prior to the long-term conflict [see, for example, 9], many informants described a lack of personal experience in pig production until the past few years and also described how this livelihood activity had become increasingly popular after people returned from the IDP camps. Pigs were mainly kept for their monetary value and produced for sale. They were generally appreciated for producing many piglets, growing fast with small inputs, and generating more income than poultry and goats. Due to the perceived high costs of building a pigsty, the majority of pigs were tethered or free roaming. Only a few informants had constructed pigsties. Some informants confined their pigs in disused mud huts, an arrangement that did not require new investment. Confinement of pigs was generally reported to reduce social tensions among community members, as free-roaming pigs often destroyed crops, which was frequently a source of conflict between smallholders. Nevertheless, some informants claimed that the lack of fresh air in the mud huts reduced the growth of their pigs and they therefore preferred them to be free roaming.

Informants commonly reported that they felt less confident dealing with animal health issues in pigs compared with the other animals they kept. A common view among the informants was also that pigs were more sensitive than other livestock, and therefore more difficult to keep healthy, as illustrated in the following quote by smallholder Gloria (woman, individual interview):

> "It has been four years now since I started with pigs. I saw that my neighbours were keeping them; I had no experience other than seeing them keeping pigs. When I was a child, no one had pigs, I'd never heard about pigs at that time. I keep pigs for money, but they bring more problems with disease than other animals, or at least it is more difficult for me to solve diseases in pigs."

The experienced sensitivity of pigs and the difficulty in treating them led to frustration and a sense of insecurity, and made informants question their own knowledge and skills in livestock production, as exemplified by smallholder Joyce (woman, individual interview):

> "If my animals die, if bad things like that happen, I feel less like a real farmer, it means a lot of struggle for me."

The informants' ongoing search for more efficient methods to deal with pig health issues can be interpreted here as stemming from a genuine concern for their animals' wellbeing. In this context, being able to ensure their animals' health was closely tied to the informants' sense of "craftmanship" in farming [see also 25]. More efficient ways of tackling animal health issues were also directly connected to the possibility of earning an income from pigs. The section below explores in more detail how the informants perceived and acted on different disease syndromes and health issues in pigs.

### Perceptions and experiences of pig diseases

The uncertainty that informants felt regarding disease spread and causes of death in pigs were often expressed along similar lines to the comments made by smallholders Peter and Beatrice (man and woman, respectively, individual interviews):

> *"I think that the pig is the most challenging animal to keep; pigs are really difficult to keep healthy. I struggle to take care of my pigs, to take control over them in a good way, to ensure that they will not get sick and to find a way to treat them when they do get sick."* (Peter)

> *"There are lots of problems with pigs getting poisoned around here, it causes them to die. Some people kill pigs, someone might feed them raw food stuff or unprepared simsim* 👁️ses-ame] *and they can die when they eat that. The problem is that it becomes difficult to know if the pigs were killed by disease or by poison, you can't be sure what caused them to die."* (Beatrice)

Some explanations among the informants concerning the spread of disease and causes of death indicated the broader framing of problems in relation to animal health, with disease not clearly separated from issues related to witchcraft or poisoning. The risk of having your pig poisoned by a community member was considered a real threat in the study area, and many informants found it difficult to distinguish between a poisoned pig and a pig suffering from an infectious disease. Mention of witchcraft and curses among informants indicated the spiritual dimension of conceptualising disease, which has also been described in other contexts [see, for example, 24 p.55].

While informants commonly reported that they had observed some clinical signs before their pigs died, they reported high levels of uncertainty regarding the causes of disease, which affected opportunities to prevent the spread of infections. The following response from smallholder Christine (woman, individual interview) sheds light on this issue:

> *"When the pigs got sick, I recognised that they were not behaving normally, something was wrong in their heads. What I mean is that they started to run around, then they were just dead, all dead on the ground. I'm not sure why they were running around. But it was like they were not stable on their legs, I could see they weren't walking properly anymore. When they died, we couldn't eat the meat since we didn't know what was wrong when the pigs died, so we threw the dead pigs into the bush."*

One reason for the perceived difficulty in interpreting disease in pigs was that practical know-how in livestock production, often passed down from parents and relatives, was mainly developed based on experiences with animal health issues in goats, poultry and cattle. This practical knowledge was sometimes found to be irrelevant or even harmful when applied to pigs, indicating the extent of how context-specific this local knowledge is and thus the difficulty of transferring it from one animal species to another [see also 20, 25, 26]. The section

below examines some of the most common pig diseases and how they were dealt with by the informants.

## Common pig diseases and treatments

In both focus group discussions and individual interviews, several informants said that pig diseases (including disease outbreaks) were more common between December and March. This correlated with the dry season, when many of them struggled to provide enough feed and water for their pigs, and therefore had to let them out to scavenge even if they had an enclosure. Other informants suggested that time of year did not have a major impact on the total occurrence of diseases, as the rainy season was perceived to increase the risk of ticks, lice and coughs, for example.

While all diseases were a potential threat to the success of pig production, some diseases were discussed as not necessarily resulting in rapid deaths, and were therefore perceived as less of a risk. Commonly mentioned pig health problems (many of them being clinical signs that were referred to by smallholders as separate diseases) that were possible to treat or did not result in rapid deaths included coughs, diarrhoea and jiggers (Table 1).

Informants had very limited access to pharmaceuticals and veterinary services. Several of those who had consulted an animal health service provider reported the poor quality of their services and advice. A variety of actors provided animal healthcare in the study area. There were veterinary officers who have a degree in veterinary medicine and look after large areas and who were therefore generally very inaccessible to the informants. The majority of informants reported that they did not know who their local veterinary officer was or how to contact them. Therefore, when facing animal health issues in livestock production, very few smallholders were able to receive assistance from a veterinary officer. Paraprofessionals with varying levels of training in animal health were generally the actors who more commonly provided advice to smallholders in the study area, and the informants generally perceived them to be veterinarians as well. Smallholders described paraprofessionals as more affordable and accessible. At the time of fieldwork in 2019, at least one paraprofessional lived in village A and several paraprofessionals lived in village B. Nevertheless, the majority of the informants relied mainly on locally available resources and the knowledge of more experienced peers for treating sick animals. This is illustrated in the response of Maria (woman, individual interview) here:

> *"When the sickness comes and you can't identify what the problem is, and you find that the pigs start to die, that's when you're supposed to call the vet doctor ☙veterinarian]. The vet doctors that are supposed to move from home to home. But mostly, you get advice from people around instead, people that have been keeping pigs for a longer time than you."*

Homemade medicine mixes initially developed to treat diseases in poultry and goat production were commonly used (Table 1). An experience repeatedly expressed, however, was that such methods seemed less efficient in pigs than in other livestock. This resonates with previous studies on farmers' local knowledge, in which evaluation, experimentation and constant adaptation are crucial for this practical knowledge to become precise enough to solve the problem at hand [20, 25, 26].

## Perceptions and experiences of ASF

In contrast to the syndromes presented in the previous section, there were also pig diseases that were seemingly impossible to deal with. Situations that were particularly difficult to

**Table 1. Description of common disease syndromes in pigs and smallholders' suggestions of how to deal with these (based on data from focus group discussions and individual interviews).**

| Pig health problem | Comments | Suggested treatments or preventive measures |
|---|---|---|
| Coughs | Perceived as a larger problem during the rainy season, and described as similar to humans having a cough. | • Marijuana leaves.<br>• Mix ash and water.<br>• Pharmaceuticals from drug shop/veterinarian. |
| Diarrhoea and vomiting | Diarrhoea was reported to be a frequent problem with goats too, and it was common to use the same treatment methods. However, these methods were described as less efficient for pigs. | • Mix salt and water.<br>• Mix leaves from local trees, washing powder and water.<br>• Pharmaceuticals (deworming or other treatment) from drug shop/veterinarian. |
| Feed intake disease | Feeding pigs raw food, such as cassava or red pepper, was described to cause sickness and skin colour changes, and in the worst case scenarios even lead to rapid deaths. | Boil cassava, maize bran and other food before feeding pigs. |
| Foot and mouth disease | Reported to be less common in pigs than cattle and said to be caused by drinking water, transmissible to other pigs and as a viral disease. | • Confine pigs.<br>• Avoid intermingling by separating pigs into different housing. |
| Heat stress | Noted as a common problem during the dry season. Pigs look tired and skin appears oily (described as "skin is melting in the sun"). | Make a hole in the ground, pour water in the hole and let pigs get in to cool down. |
| Jiggers[a] | Described as a disease that enters through pigs' feet, making the pigs' legs unstable, and that can be spread to humans. | • If pigs are confined, regularly smear the floor or ground with soil and cold water to kill and prevent jiggers.<br>• Pharmaceuticals from drug shop/veterinarian. |
| Runny nose | Described as sweat from the nose [snout]. It was reported that it was hard to prevent pigs with a runny nose and that were also weak from dying. | • Same treatment as cough.<br>• Pharmaceuticals from drug shop/veterinarian. |
| Scabies/lumps | Described as causing spots, sores or marks on the pigs' body, making pigs thin, and can cause death within a month if efficient treatment not found. | Avoid keeping pigs in a wet and/or muddy place. |
| Swollen stomach | Reported to be due to feed intake or worms. | • Mix washing powder and water to prevent and treat.<br>• Mix salt, washing powder and water to prevent and treat.<br>• Pharmaceuticals (deworming) from drug shop/veterinarian. |
| Ticks and lice | Described as difficult to discover since they are so small. | • Avoid keeping pigs in wet and/or muddy places.<br>• Wash pigs with water.<br>• Spray with pesticides.<br>• Use paraffin as a treatment. |

[a]Jiggers is a parasitic insect. Infection occurs due to penetration of the female sand flea (*Tunga penetrans*) into the skin of humans and animals, usually attacking hands or feet. Infection can be recognised through bumps under the skin. Jiggers infection often causes intense itching, followed by inflammation and acute pain [42].

handle were when pigs died rapidly after the first signs of sickness, as Nancy (woman, individual interview) highlights here:

> "A big problem with pigs is that they are very weak and get a lot of sickness. Think about goats, they are stronger and can be sick for longer before they die. So, when you have pigs and realise something is wrong, it becomes very difficult. They can die after just a few days, and then you do not even have time to see what was wrong, what made them sick in the first place."

Like Nancy, several informants had experiences of rapid death in pigs. Some of them had given up on pig production as a result. Smallholder Morris (man, individual interview) describes such an experience here:

> "*Pigs get more diseases than other animals. Some very difficult ones. One challenge I have had with pigs has been visible during the months of January and February. The signs I could see was that the pigs stopped moving as before, they stopped eating, they were just lying down, like two days they were sick and then they just died. When they started to get sick, there was sweat from the nose, but I don't know the name of this disease. I decided to not have more pigs after this experience because I would not know how to solve this disease if it happened again with some new pigs.*"

Few informants explicitly talked about ASF. Instead, they often used a variety of names to describe similar syndromes, where the general theme was that several pigs were affected at the same time and that it was difficult to prevent the pigs from dying, despite different attempts to treat them. Based on knowledge of ASF epidemiology in East Africa [see, for example, 43] and documented outbreaks of ASF in the study area [44], this group of similar syndromes was interpreted by us as being descriptions of experiences with ASF. However, this should not be taken to mean that informants see these syndromes as being the same disease or as stemming from the same disease-causing agent (Table 2).

## Local responses to different syndromes interpreted as ASF

There was limited local understanding of the fact that there is currently no cure for ASF. Instead, when faced with syndromes that were interpreted by us to be ASF and that caused the rapid death of pigs, informants tended to interpret this as a result of not having access to the correct advice or treatment, as described by Charles (man, individual interview):

> "*Sometimes there is orere [disease outbreaks] in the pigs, they all get sick at once, and then a lot of pigs can die without us being able to do anything about it. That's a big problem. Since the vets are almost never here, or perhaps only like two times per year to give some vaccines, we can't get much help from them. We can't rely on the vets to keep our animals healthy.*"

Informants had experience of the effectiveness of purchased medication to treat some clinical signs, such as diarrhoea, and sometimes they pooled resources to fund one person's travel costs to purchase pharmaceuticals in town. The informants' use or aspiration to use pharmaceuticals is an example of the fluid relation between different technologies and knowledge systems in this study context [22]. Opinions varied as to whether pharmaceuticals would be efficient for treating the syndromes that were interpreted by the authors to be ASF (Table 2). For smallholder Blenda (woman, individual interview), the question was not whether pharmaceuticals would be useful in the case of ASF, but that the perceived difficulties were instead related to a lack of access and knowing what specific pharmaceuticals would be efficient in this context:

> "*With pigs, I don't know much about sickness in pigs. But they just started dying. Their bodies became very thin. There was a lot of saliva from their mouths. How to know which drugs to give them? And how to get the drugs? They just die.*"

The findings from the interviews and focus group discussions show that many informants had experience of "orere" in poultry, possibly caused by Newcastle disease (ND). Vaccination

**Table 2. Local description of syndromes that the authors interpret as representing African swine fever (ASF), descriptions of clinical signs, and treatment and prevention method used (based on data from focus group discussions and individual interviews).**

| Local names of ASF | Descriptions | Local treatment and prevention method | Comments |
|---|---|---|---|
| African swine fever | Weak body, saliva from the mouth, sleepy, sneezing, sweat from ears and nose, rapid death, colour changes in bones. | • Regular deworming of pigs was reported to reduce the risk of ASF.<br>• Keeping pigs in the same place, thus avoiding pigs intermingling with other people's pigs, was said to prevent ASF.<br>• Avoiding bringing meat from the pork joint in the village centre back home was described as preventing ASF.<br>• Stop pigs eating the bones of dead pigs, as ASF was said to be stuck in bones.<br>• Injections or pharmaceuticals from veterinarians were suggested as a measure to prevent and control ASF.<br>• Described as difficult to prevent and treat. | Uncertainty among informants about the efficiency of injections in the case of ASF. |
| Malaria | Weakness, colour changes behind the ears and in the skin (darker), shaking body, sweat from the nose, saliva from the mouth, loss of appetite, sleepy, sneezing, changed colour of meat, rapid death of all pigs. | • Consulting a veterinarian to deworm the infected pigs was suggested to prevent malaria.<br>• Described as difficult to prevent and treat. | • It was reported that the health of pigs only appeared to improve temporarily after deworming.<br>• Several informants believed that malaria was treatable, but the problem was that they did not have any efficient treatment to hand. |
| Orere[a]/ outbreaks | Body becomes weak and thin, hair standing up, saliva from mouth, dark spots on body, pigs running around in circles, tail hanging down, loss of appetite, vomiting, colour changes in meat, affecting several pigs at once, rapid death. | • Deworming believed to reduce risks of infection.<br>• Calling veterinarian to get an injection (type of injection unknown/unspecified) was believed to reduce the risk of infection and also enhance chances of controlling outbreaks.<br>• Treat with papaya leaves.<br>• Treat with mix of washing powder and water.<br>• Improvement of general conditions for pig keeping was described as important to avoid infection: feed, regularly provide water, confine pigs (confinement of pigs to avoid infection).<br>• Described as difficult to prevent and treat. | • One informant believed that the colour of meat did not change because of the disease, but due to being treated with a mix of washing powder and water, and another that the colour of meat changed due to treatment with papaya leaves.<br>• Administering injections (as fast as possible after recognising infection) was described as potentially helpful, but it was also reported that the health only appeared to improve temporarily.<br>• Many informants described how it was difficult to construct housing due to financial constraints. |
| Running around | Changed behaviour, running around in circles, unstable legs, loss of energy, rapid death. | Described as difficult to prevent and treat. | |

[a]Orere (disease outbreaks) was not a term restricted to disease outbreaks in pigs, but was also used when describing disease outbreaks in poultry. Disease outbreaks were referred to by smallholders in both English and Luo.

was commonly suggested as the preferable prevention and treatment method in this instance. Due to financial constraints and limited access to vaccines and veterinary services, few were able to vaccinate against ND. This might partly explain why some of the informants believed that "orere" in pigs could also be prevented with vaccinations.

Results also show that some of the informants who believed that all kinds of syndromes in pigs could be treated and cured with pharmaceuticals identified time as one of the most critical factors in successful treatment, as expressed by smallholder David (man, FGD):

> "There is a disease that we call orere. If that disease comes, you will find that the pigs are running around the compound. They can also become weak and usually they die, all of them at once. But you can consult a vet, and then the disease can get cured if you just get some help from veterinarians. But this one, with the orere, if it stays for too long in the pigs, it can be hard to cure. So you have to get a vet to cure it very fast for the injection to help. In this sense, there is no disease that can't get cured; everything can be solved with drugs."

Some informants described how they had consulted an animal health service provider as a last resort when they were unable to deal with ASF (as well as other pig diseases) themselves. The person had then injected or dewormed the pigs, informing them that this would make the disease disappear. In this sense, responding to ASF also included elements of discerning between different, and sometimes conflicting, information, a situation that could lead to confusion about what knowledge to trust or not [see also 14]. In the following quote, smallholder Margret (woman, FGD) describes such a scenario:

*"I experienced a difficult disease in my pigs, it was when the malaria came. The pigs began to sweat and their ears were filled with blood. One time when that happened to me, I called a vet. He gave some deworming to my pigs; he told me it was the worms that gave the weakness to my pigs, that disturbed them. My pigs first seemed to improve a bit, but then they died from this malaria, they all died very quickly."*

Overall, suggestions about how to prevent and treat what we here interpret as ASF differed between the informants. This was largely linked to their perceptions about how this disease spreads. A limited number of informants were well aware that it is not possible to treat ASF, and understood that biosecurity measures are needed to prevent it spreading before the animals get ill. This was evident, for example, in a response by smallholder Evelyn (woman, FGD):

*"But to prevent ⬤ASF], I think you should keep them in the same place and also not bring meat back home from the pork place in the centre. There was ASF here, about two months ago, it was here and my pigs died because of ASF. They ate the bones of dead pigs and died soon after."*

Several informants also emphasised that ASF can be transmitted to healthy pigs that are in contact with dead infected pigs or contaminated material, and therefore stressed that dead pigs should ideally be buried or burnt to avoid other pigs getting infected. Very few informants, if any however, implemented such measures. Instead, many butchered the dead pigs at home and sold the pork to community members. If the meat appeared unpleasant, dead pigs were reportedly thrown in the bush or used as dog feed. A common response when realising that something might be wrong with their pigs and fearing that money will be lost if the pigs die, was to sell the pigs, as described in smallholder Judith's (woman, individual interview) response here:

*"Last year I had four pigs, but I sold them to pay the school fees for my children. I haven't bought new ones. It was difficult to keep pigs because the neighbour complained a lot about crops getting destroyed. But they got sick also. I could tell that something was wrong when looking at them. They started vomiting and got diarrhoea. One time, I called the doctor for animals, and someone came here and treated the pigs. They did not fully improve, but at least enough so that I could sell them."*

A few informants reported a reluctance to approach veterinary actors in relation to ASF. For example, one informant had heard on the radio that ASF outbreaks should be reported to the District Veterinary Officer, but was fearful that such reporting would lead to a request for all animals to be culled or to the implementation of other biosecurity measures requiring financial investment. This reflects the importance of acknowledging local conditions that affect smallholders' ways of knowing and responding to pig diseases such as ASF when considering

how veterinary knowledge and preventive measures to avoid ASF infection might have relevance in the local context [see also 36].

## Discussion

As mentioned earlier in this paper, several scholars have confirmed in empirical studies that smallholders' practical knowledge can play a key role in solving problems in the local context with regards to animal disease [see, for example, 19]. It has also been acknowledged that smallholders rather than scientists often are the main producers of locally relevant novelties in agriculture, illustrating the value and importance of this context-specific, dynamic and experimental local knowledge [20, 25, 26]. These aspects are important to underline, not least in light of past and current tendencies to ignore and suppress the skills and knowledge of smallholders in the face of hegemonic Western science and colonialism [19, 45]. What the results from the present study show, however, is that the smallholders' practical knowledge was not enough to solve pig health issues adequately, which points to the need to combine métis and scientific knowledge in this context.

### The limitations of practical knowledge in controlling pig diseases

The results show that compared with other livestock in the study area, pigs were generally perceived to be more sensitive and harder to keep healthy. In dealing with pig health issues, the informants mainly relied on the resources and knowledge accessible to them in their local communities. In relation to this, they often reported that they lacked the means to diagnose and treat sick pigs, something that has also been described in other studies on Ugandan smallholder pig production [see, for example, 14]. The smallholders' strong dependence on local knowledge in pig production can partly be understood as a consequence of the limited access and sometimes poor quality of veterinary services available in the study area. Their request for other kinds of knowledge in dealing with pig health issues indicates the limitation of practical knowledge as a means of controlling ASF and other pig diseases [see also 16, 22]. The practical know-how and treatment methods used in pig production had mainly been developed in relation to poultry, goats and cattle, and were commonly perceived as less efficient in pigs. Thus, the relatively short history of pig production in the study setting can be assumed to play a key role in this perceived difficulty. This resonates well with how practical knowledge has been described and theorised in the literature, where it is assumed to develop over time through the constant adaptation, experimentation and fine-tuning of methods [20, 25, 26].

### Local perceptions and responses to pig health issues

The informants reported that all pig health issues presented potential risks to their pig production. ASF and descriptions of syndromes and diseases that the authors interpreted as ASF were commonly described as being particularly difficult and stressful to handle, not least because it could cause the rapid death of all pigs. It is important to emphasise, both with regards to ASF as well as other pig health issues mentioned in this paper, that the disease terms used by the informants generally related to syndromes and not to specific diagnoses [see also 19, 22, 23].

Many informants expressed uncertainties about how ASF is transmitted, which may partly explain why they found it so difficult to prevent and control this disease. The complexity of local pig health should also be stressed here. This is related to previous reports about pigs and other livestock in Uganda often suffering from undernourishment as well as several subclinical infections at once [15, 46]. Assuming that this could also be the case in the context studied here, this may be a potential explanation for why several informants found it hard to distinguish ASF from other pig diseases. This study also identified uncertainties and different views

among the informants regarding whether ASF was curable or not. Some of the informants, believing it possible to cure ASF, described the main hindrance to be a lack of access to pharmaceuticals and efficient treatment methods. This sheds light on the smallholders' search for more accurate solutions in dealing with pig health issues, as well as their openness to combining their practical knowledge with aspects of scientific knowledge that have also been revealed in other studies on smallholder livestock production in sub-Saharan Africa [see, for example, 22, 23]. Some informants who had been in contact with an animal health service provider reported how this openness could entail risks, such as receiving incorrect advice or erroneous information in relation to ASF. This has also been reported in other studies on Ugandan smallholder pig production [14, 47, 48]. In this sense, the smallholders were not only at risk of losing money based on poor advice from animal health service providers, but incorrect information could potentially also exacerbate a general sense of uncertainty about how to act when pigs are infected by incurable diseases such as ASF.

The results from this study show that very few smallholders implemented preventive measures to hinder the spread of AFS. Preventive measures commonly recommended to farmers include constructing pigsties, buying an extra pair of boots to be used when entering the pigsty, and using commercially produced disinfectants to prevent the spread of ASF [see, for example, 49, 50]. With limited financial resources, however, the majority of informants considered preventive measures, such as the construction of pigsties, to be too costly. It was also a common practice among them to sell sick pigs as a strategy to reduce or avoid financial losses in the event of pig diseases. In such cases, the informants generally perceived it to more critical for their pigs to be healthy enough to sell rather than to resolve the actual sickness. In this sense, responses to pig health issues including ASF should not only be understood in the light of what knowledge the informants have, but can also be seen as a pragmatic response to having to meet numerous household needs and thus trying to make the best of a difficult situation. At the same time, the fact that some informants reached out to animal health service providers in an attempt to complement their practical know-how in pig production reveals how animal health was still very central to their pig production. The findings from this study further show that the informants generally made minimal investments in their pig production. Against this backdrop, the authors agree that there is a need to move beyond "universal" preventive approaches in relation to ASF [8], and encourage explorations of options and solutions better suited to poverty-constrained smallholder contexts [see also 51]. This is believed to have more relevance in the study setting, where the informants generally perceived commercial products and the construction of pigsties to be too expensive.

## Barriers to combining knowledge systems

When attempting to combine knowledge systems in order to deal with animal health issues more efficiently, key factors for consideration have been discussed in several empirical studies on smallholder livestock production in the Global South [see, for example, 19, 22, 24]. One such aspect concerns different perspectives on health and disease between veterinary practitioners and smallholders. For example, a study from Kenya shows how the Western science used by veterinarians is based on a view of health as the normal state, in contrast to disease being something abnormal [19]. In that study, veterinarians assumed medical treatments to be required to return the livestock body to normality, whereas pastoralists did not differentiate conceptually or spatially between disease and health, but perceived diseases to be a natural part of the environment [19]. Thus, pastoralists considered treatment as potentially required to reduce the unnecessary loss of cattle, but did not perceive it as an imperative in order to eradicate or avoid diseases [19]. Similar observations were made in a study from western Uganda

where cattle-keeping farmers perceived minor health issues and poor growth as something normal, rather than something worth controlling, despite the negative effects this had on their income and food security [15]. Another study on Ugandan pig farming showed that smallholders were more concerned with the growth of their pigs and their pigs appearing to be in good health so that they could sell them, rather than preventing the spread of ASF, despite this being a top priority for veterinarians [8]. Another interesting finding from that study was how smallholders generally preferred their pigs to be free roaming, partly due to their perception of pigs as part of the household and therefore not separable from humans through confinement [8]. These examples illustrate how priorities, methods and epistemologies may differ between animal health service providers and smallholders when it comes to animal health and disease [see also 52]. With this in mind, improved understanding of smallholders' ways of knowing and conceptualising diseases will be critical for improving communication between smallholders and veterinarians, and ensuring appropriate animal health service delivery [22, 23].

## Conclusions

Since ASF cannot be treated or cured, the only available option to reduce the negative impact of this disease is prevention and control. The findings from this study reveal that the opportunities and motivations among smallholders for implementing preventive measures in pig production were generally low. Overall, the informants acted once they recognised visible signs of sickness in their pigs, indicating how the concept of prevention was not obvious in this study context. As mentioned above, the responses among informants to realising that something might be wrong with their pigs were not always concentrated on preventing the further spread of diseases. Thus, if the local conditions for pig production and smallholders' ways of understanding are not given greater acknowledgment, there are reasons to believe that the strategies for prevention and control, as suggested by researchers and veterinary services, will be difficult to implement in smallholder contexts. In relation to this, it may be relevant to explore opportunities to recommend preventive and controlling measures in relation to ASF that could be motivated by factors other than disease [53, 54]. These include how the confinement of pigs in the study context can potentially be perceived as more relevant–for reducing social tensions in the community due to crops being destroyed by free-roaming pigs–than preventing and controlling pig diseases, which is often the main focus of veterinarians. In so doing, there is potential to improve the communication between smallholders and animal health providers by acknowledging smallholders' needs and wants in their pig production. At the same time this could also help reduce the negative impacts of ASF and other pig health issues.

Achieving more efficient control of ASF in Uganda is not only important for reducing the negative socio-economic impacts of this disease [44] but, from what has been seen in this study, can also be important for increasing smallholders' motivations to continue keeping pigs. As mentioned earlier in this paper, the informants perceived pig production as having the potential to provide an income and thus enhance their opportunities in life, which was generally the main motivation for starting pig production in the first place. At the same time, ASF and other pig health issues have forced some of them to abandon pig production altogether, as they saw no possibility of dealing with these issues if they reappeared in new pigs. In order to boost the potential of small-scale pig production as a poverty mitigation strategy in Uganda, it is suggested that the structural factors influencing local conditions for pig production should be addressed, such as increasing smallholders' access to adequate animal health services. Indeed, the informants requested complementary knowledge about how to deal with major and minor pig health issues. Nevertheless, for veterinary advice and knowledge to have

relevance in this and similar contexts, veterinary actors and researchers need to pay careful attention to smallholders' problem framings and ways of knowing in livestock production.

## Acknowledgments

We are grateful to all the smallholder farmers who participated in this study, sharing their time and rich knowledge in livestock production with us. We would also like to thank the field assistants, Alfred Ojok and Auma Susan Obol, for all their assistance during and after fieldwork. Tonny Aliro, Peter Ogweng and Charles Masembe also provided invaluable project support.

## Author Contributions

**Conceptualization:** Anna Arvidsson, Klara Fischer, Erika Chenais, Susanna Sternberg-Lewerin, Karl Ståhl.

**Formal analysis:** Anna Arvidsson.

**Funding acquisition:** Klara Fischer, Erika Chenais, Juliet Kiguli, Susanna Sternberg-Lewerin, Karl Ståhl.

**Investigation:** Anna Arvidsson.

**Methodology:** Anna Arvidsson, Klara Fischer, Erika Chenais, Susanna Sternberg-Lewerin.

**Project administration:** Juliet Kiguli, Karl Ståhl.

**Supervision:** Klara Fischer, Erika Chenais, Juliet Kiguli, Susanna Sternberg-Lewerin, Karl Ståhl.

**Writing – original draft:** Anna Arvidsson.

**Writing – review & editing:** Klara Fischer, Erika Chenais, Juliet Kiguli, Susanna Sternberg-Lewerin, Karl Ståhl.

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
