## [Decision Letter · Decision Letter 0]

20 Feb 2023

PONE-D-23-01026Limitations and opportunities of smallholders’ practical knowledge when dealing with pig health issues in northern UgandaPLOS ONE

Dear Dr. Arvidsson,

Thank you for submitting your manuscript to PLOS ONE. After careful consideration, we feel that it has merit but does not fully meet PLOS ONE’s publication criteria as it currently stands. Therefore, we invite you to submit a revised version of the manuscript that addresses the points raised during the review process.Check the reviewers' report below and respond to their comments. Clarify the availability of veterinary services in the study area.Clarify if the research team has triangulated some of the findings with local veterinary services

More details on the farmers' experience rearing pigs can help to understand why they have poor knowledge about pig diseases and their management. For example, in neighboring countries (Kenya, Rwanda), pig rearing is perceived by smallholders as easier compared to other type of livestock production. There need to be more explanations on why the situation is different in the study area. 

We look forward to receiving your revised manuscript.

Kind regards,

Anselme Shyaka, Ph.D

Academic Editor

PLOS ONE

Journal Requirements:

" ext-link-type="uri" xlink:type="simple">https://journals.plos.org/plosone/s/file?id=ba62/PLOSOne_formatting_sample_title_authors_affiliations.pdf"

2. In the ethics statement in the Methods, you have specified that verbal consent was obtained. Please provide additional details regarding how this consent was documented and witnessed, and state whether this was approved by the IRB.

Additional Editor Comments:

Thanks for submitting your paper for consideration by PLOS ONE.

Reviewers have recommended to revise the minor comments in each reviewer's report.

In addition, I advise you to provide more details on the availability of veterinary services in the study area as well as the study participants' expertise in pig rearing. The two points could contribute to further understanding of :the lack of knowledge in pig diseases and pig diseases managementavailable veterinary expertise in the study setting. This could help understanding the reason behind the reported possible misinformation from animal health professionals as well as the farmers' lack of trust in veterinary services.

Reviewers' comments:

Reviewer's Responses to Questions

**Comments to the Author**

1. Is the manuscript technically sound, and do the data support the conclusions?

Reviewer #1: Yes

Reviewer #2: Yes

2. Has the statistical analysis been performed appropriately and rigorously? 

Reviewer #1: Yes

Reviewer #2: N/A

3. Have the authors made all data underlying the findings in their manuscript fully available?

Reviewer #1: Yes

Reviewer #2: Yes

4. Is the manuscript presented in an intelligible fashion and written in standard English?

Reviewer #1: Yes

Reviewer #2: Yes

5. Review Comments to the Author

Reviewer #1: A general comment: The authors should have crosschecked information with the veterinary authorities in Uganda.

Reviewer #2: I have a couple of minor comments below.

L254 : how recent is “more recently” maybe give a year or a range?

L281: Would be good to mention the sex/gender for the quote and the name of the village in the whole text

6. PLOS authors have the option to publish the peer review history of their article (what does this mean?). If published, this will include your full peer review and any attached files.

Reviewer #1: No

Reviewer #2: **Yes: **Michel Dione

---

## [Author Response · Author response to Decision Letter 0]

27 Apr 2023

Editor: We have incorporated all of your suggestions into our revision. They were very informative. Thank you for all the helpful comments.

Reviewer 1: We have incorporated all of your suggestions into our revision. They were very insightful. Thank you.

Reviewer 2: We have incorporated all of your suggestions, except the suggestion to state the names of the study villages due to protecting the anonymity of the informants, into our revision. Thank you for the constructive comments.

---

## [Decision Letter · Decision Letter 1]

30 May 2023

Limitations and opportunities of smallholders’ practical knowledge when dealing with pig health issues in northern Uganda

PONE-D-23-01026R1

Dear Ms. Arvidsson,

We’re pleased to inform you that your manuscript has been judged scientifically suitable for publication and will be formally accepted for publication once it meets all outstanding technical requirements.

Kind regards,

Anselme Shyaka, Ph.D

Academic Editor

PLOS ONE

Additional Editor Comments (optional):

Reviewers' comments:

Reviewer's Responses to Questions

**Comments to the Author**

1. If the authors have adequately addressed your comments raised in a previous round of review and you feel that this manuscript is now acceptable for publication, you may indicate that here to bypass the “Comments to the Author” section, enter your conflict of interest statement in the “Confidential to Editor” section, and submit your "Accept" recommendation.

Reviewer #1: All comments have been addressed

Reviewer #2: All comments have been addressed

2. Is the manuscript technically sound, and do the data support the conclusions?

Reviewer #1: Yes

Reviewer #2: Yes

3. Has the statistical analysis been performed appropriately and rigorously? 

Reviewer #1: Yes

Reviewer #2: Yes

4. Have the authors made all data underlying the findings in their manuscript fully available?

Reviewer #1: Yes

Reviewer #2: (No Response)

5. Is the manuscript presented in an intelligible fashion and written in standard English?

Reviewer #1: Yes

Reviewer #2: Yes

6. Review Comments to the Author

Reviewer #1: I had a re-look and all the comments were addressed. .

Reviewer #2: (No Response)

7. PLOS authors have the option to publish the peer review history of their article (what does this mean?). If published, this will include your full peer review and any attached files.

Reviewer #1: No

Reviewer #2: **Yes: **Michel Mainack Dione

---

## [Editor Report · Acceptance letter]

1 Jun 2023

PONE-D-23-01026R1 

Limitations and opportunities of smallholders’ practical knowledge when dealing with pig health issues in northern Uganda 

Dear Dr. Arvidsson:

I'm pleased to inform you that your manuscript has been deemed suitable for publication in PLOS ONE. Congratulations! Your manuscript is now with our production department. 

Kind regards, 

on behalf of

Dr. Anselme Shyaka 

Academic Editor

PLOS ONE